# Genetically Raised Circulating Bilirubin Levels and Risk of Ten Cancers: A Mendelian Randomization Study

**DOI:** 10.3390/cells10020394

**Published:** 2021-02-15

**Authors:** Nazlisadat Seyed Khoei, Robert Carreras-Torres, Neil Murphy, Marc J. Gunter, Paul Brennan, Karl Smith-Byrne, Daniela Mariosa, James Mckay, Tracy A. O’Mara, Ruth Jarrett, Henrik Hjalgrim, Karin E. Smedby, Wendy Cozen, Kenan Onel, Arjan Diepstra, Karl-Heinz Wagner, Heinz Freisling

**Affiliations:** 1Department of Nutritional Sciences, Faculty of Life Sciences, University of Vienna, 1090 Vienna, Austria; nazlisadat.seyedkhoei@univie.ac.at (N.S.K.); karl-heinz.wagner@univie.ac.at (K.-H.W.); 2Colorectal Cancer Group, ONCOBELL Program, Bellvitge Biomedical Research Institute (IDIBELL). L’Hospitalet de Llobregat, 8908 Barcelona, Spain; rcarreras@idibell.cat; 3Nutrition and Metabolism Branch, International Agency for Research on Cancer (IARC-WHO), 69008 Lyon, France; MurphyN@iarc.fr (N.M.); gunterm@iarc.fr (M.J.G.); 4Genomic Epidemiology Branch, International Agency for Research on Cancer (IARC-WHO), 69008 Lyon, France; BrennanP@iarc.fr (P.B.); smith-byrnek@fellows.iarc.fr (K.S.-B.); mariosad@iarc.fr (D.M.); mckayj@iarc.fr (J.M.); 5Department of Genetics and Computational Biology, QIMR Berghofer Medical Research Institute, 4006 Brisbane, Australia; 6Institute of Infection, Immunity and Inflammation, MRC-University of Glasgow Centre for Virus Research, Glasgow G61 1QH, UK; Ruth.jarrett@glasgow.ac.uk; 7Department of Epidemiology Research, Statens Serum Institut, 2300 Copenhagen, Denmark; HHJ@ssi.dk; 8Department of Hematology, Finsen Centre, 2100 Copenhagen, Denmark; 9Department of Medicine Solna, Division of Clinical Epidemiology, Karolinska Institutet, 171 77 Stockholm, Sweden; Karin.Ekstrom.Smedby@ki.se; 10Department of Hematology, Karolinska University Hospital, S-141 86 Stockholm, Sweden; 11Departments of Preventive Medicine and Pathology, Keck School of Medicine of USC, University of Southern California, Los Angeles, CA 90033, USA; wcozen@usc.edu; 12Department of Genetics and Genomics Sciences, Icahn School of Medicine at Mount Sinai, New York, NY 60637, USA; kenan.onel@mssm.edu; 13Department of Pathology and Medical Biology, University of Groningen, University Medical Center Groningen, 9713 Groningen, The Netherlands; a.diepstra@umcg.nl

**Keywords:** Bilirubin, UGT1A1, Mendelian randomization, cancer risk

## Abstract

Bilirubin, an endogenous antioxidant, may play a protective role in cancer development. We applied two-sample Mendelian randomization to investigate whether genetically raised bilirubin levels are causally associated with the risk of ten cancers (pancreas, kidney, endometrium, ovary, breast, prostate, lung, Hodgkin’s lymphoma, melanoma, and neuroblastoma). The number of cases and their matched controls of European descent ranged from 122,977 and 105,974 for breast cancer to 1200 and 6417 for Hodgkin’s lymphoma, respectively. A total of 115 single-nucleotide polymorphisms (SNPs) associated (*p* < 5 × 10^−8^) with circulating total bilirubin, extracted from a genome-wide association study in the UK Biobank, were used as instrumental variables. One SNP (rs6431625) in the promoter region of the uridine-diphosphoglucuronate glucuronosyltransferase1A1 (*UGT1A1)* gene explained 16.9% and the remaining 114 SNPs (non-*UGT1A1* SNPs) explained 3.1% of phenotypic variance in circulating bilirubin levels. A one-standarddeviation increment in circulating bilirubin (≈ 4.4 µmol/L), predicted by non-*UGT1A1* SNPs, was inversely associated with risk of squamous cell lung cancer and Hodgkin’s lymphoma (odds ratio (OR) 0.85, 95% confidence interval (CI) 0. 73–0.99, *p* 0.04 and OR 0.64, 95% CI 0.42–0.99, *p* 0.04, respectively), which was confirmed after removing potential pleiotropic SNPs. In contrast, a positive association was observed with the risk of breast cancer after removing potential pleiotropic SNPs (OR 1.12, 95% CI 1.04–1.20, *p* 0.002). There was little evidence for robust associations with the other seven cancers investigated. Genetically raised bilirubin levels were inversely associated with risk of squamous cell lung cancer as well as Hodgkin’s lymphoma and positively associated with risk of breast cancer. Further studies are required to investigate the utility of bilirubin as a low-cost clinical marker to improve risk prediction for certain cancers.

## 1. Introduction

Cancer is a major cause of morbidity and mortality globally, and the number of new cancer cases is expected to increase further over the next decades (CDC, 2020). In 2018, there were over 18 million new cancer cases and nine million cancer-related deaths [1].

Cancer-promoting inflammation is an enabling characteristic of cancer development, and inflammatory cells can also release reactive oxygen species [2]. A major cause of cancer is damage to DNA as a result of oxidative stress, mainly due to excess reactive oxygen species, antioxidants depletion, or both [3].

Bilirubin, a metabolic by-product of hemoglobin breakdown, is one of the most potent endogenous antioxidants of the human body and also has substantial anti-inflammatory properties [4,5,6,7]. Therefore, bilirubin may play a preventive role in cancer development. Blood levels of bilirubin are under genetic control via expression of uridine-diphosphoglucuronate glucuronosyltransferase1A1 (UGT1A1) in the liver, which converts insoluble bilirubin into a more water-soluble form for renal and biliary excretion [8]. Individuals homozygous for seven thymine–adenine (TA)-repeats (7/7) at the *UGT1A1*28* locus have decreased enzyme activity, which leads to a less effective glucuronidation and moderately higher than normal blood levels of bilirubin (known as Gilbert’s syndrome, GS) [9,10].

The seven TA-repeats allele of *UGT1A1*28* polymorphism underlying GS was investigated in relation to cancers of the endometrium [11], ovary [12], lung [13,14], breast [15], and prostate [16]. However, results of these studies were inconclusive, did not specifically investigate bilirubin as a putative cancer risk factor, and had limited sample size (range of number of cases 129 to 310), with the exception of Horsfall et al., where an inverse association was observed between genetically raised bilirubin levels and lung cancer risk among current smokers [14]. 

In this study, we investigated whether genetically raised circulating bilirubin levels are causally associated with risk of ten cancers using a Mendelian randomization (MR) approach. This technique uses genetic variation as instrumental variables [17], and in the absence of pleiotropy, an association between the genetic instruments and the outcome implies that the risk factor of interest may have a causal role in disease etiology (here: cancer risk) [18]. MR addresses unmeasured confounding (e.g., by smoking), which is a major limitation of observational studies [17]. The ten cancer types were investigated in large international consortia and selected based on previous evidence (cancers of the lung, ovary, endometrium, breast, and prostate) [11,12,13,14,15,16] or biological plausibility (pancreatic cancer, renal cell cancer, Hodgkin’s lymphoma, melanoma, and neuroblastoma).

## 2. Materials and Methods

In a two-sample MR approach, we first identified 115 single-nucleotide polymorphisms (SNPs) that were genome-wide associated (*p* < 5 × 10^−8^) with circulating total bilirubin levels in a genome-wide association study (GWAS) that included 317,639 individuals of European ancestry (white British) from the UK Biobank (UKB) (non-British white, South Asian, African, and East Asian GWASs were excluded based on a combination of self-identification and refinement using population-specific genotype principal components) [19]. The UKB project, a long-term prospective cohort study, recruited about 500,000 people aged between 40 and 69 years in 2006–2010 from across the UK [20]. The raw total bilirubin levels were adjusted for age, sex, the top 40 principal components for population stratification, recruitment center, indicators of socioeconomic status, and potential technical confounders (blood and urine sampling time, fasting time, and sample dilution factor) [19]. SNPs were independently associated with total bilirubin levels, which were reflected by measures of linkage disequilibrium (LD R^2^ < 0.001). The SNPs with ambiguous strand codification (T-A or guanine-cytosine, G-C) were replaced by SNPs in LD R^2^ > 0.8 in European populations using the *proxysnps* R package. The summary statistics for the associations of SNP allele dosage with standardized bilirubin levels are shown in Appendix A.

One SNP (rs6431625) in the promoter region of the *UGT1A1* gene in chromosome 2 explained 16.9% of phenotypic variance in circulating total bilirubin levels, which was estimated as a function of the effect size for the risk factor in standard deviation units and the minor allele frequency [21]. This SNP was in strong linkage disequilibrium (LD R^2^ = 0.74) with the *UGT1A1*28* promoter TA-repeats polymorphism (rs3064744) [22]. The remaining 114 SNPs (non-*UGT1A1* SNPs) explained 3.1% of the total bilirubin variance [22] and provided an F-statistic for the strength of the relationship between the genetic instrument and the bilirubin levels of 89.1. The F-statistic is an estimation of the magnitude of the instrument bias (e.g., F-statistic < 10 for the weak instruments) [23].

Second, ten cancer types were analyzed using genetic data that together summed up to a total of 336,110 cancer cases and 589,467 controls of European ancestry (Table 1). Cases and controls were individually matched in the original GWAS for each cancer, including pancreatic cancer (overall and sex subgroups) [24,25,26], renal cell cancer (overall and sex subgroups) [27], lung cancer (overall, ever and never smokers subgroups, and histological subtypes of adenocarcinoma, squamous cell, and small cell) [28], ovarian cancer (overall and serous subgroup) [29], endometrial cancer [30], breast cancer (overall and estrogen receptor (ER) subgroups) [31], prostate cancer [32], Hodgkin’s lymphoma [33], melanoma, [34] and neuroblastoma [35]. To prevent weak instrument bias, these genetic data did not include samples from the UKB. Each contributing study was approved by the appropriate institutional review board/ethics committee. All participants provided informed consent. 

SNPs summary estimates (β_cancer_) were retrieved from these recently published GWAS results, which were obtained from: their cancer genetic consortia [27,28,29,30,31,32,36], the Genotypes and Phenotypes database (dbGaP) [37], public web repositories, and the MR-Base platform [38].

Imputed SNPs were restricted based on imputed accuracy, and only SNPs with high imputation quality (R^2^ > 0.8) were selected for our analyses. SNPs summary statistics for genetic associations with risk of the ten cancers are shown in Appendix A.

### Statistical Analyses

*A priori* power calculations were performed for MR associations of nominal statistical significance (α < 0.05) between both the *UGT1A1* SNP and non-*UGT1A1* SNPs, respectively, and cancer risk; given the explained variance and the sample sizes for the different cancers, using the method proposed by Burgess et al. [39].

Estimated risk effects were obtained for each genetic variant, named Wald estimate (genetic effect on cancer risk [β_cancer_]/genetic effect on total bilirubin levels [β_bilirubin_]). 

As the main MR approach used in the analyses of the large SNPs set, excluding the *UGT1A1* SNP, the Wald estimates were combined in a single estimate through a likelihood-based MR approach. This approach is considered to be the most robust under the general assumption for all MR methods regarding linear relationship between the exposure and the outcome [40], which we assumed in the range of bilirubin variation reflected by these SNPs.

The initial step in the sensitivity analyses was to apply the inverse-variance weighted (IVW) MR approach [41] and to assess the presence of outlier observations among the SNP Wald estimates using the MR pleiotropy residual sum and outlier (MR-PRESSO) test [42]. The MR-PRESSO approach identifies heterogeneity between SNP effects (*p*_Global_) as an evidence of directional horizontal pleiotropy, identifies outlier SNPs, and tests if the presence of outliers is biasing the estimation of risk (*p*_Distortion_).

To evaluate the extent to which directional pleiotropy may affect the risk estimates, we used the intercept test within a MR-Egger weighted linear regression approach [43]. Moreover, the weighted median [21] and the modal-based estimate MR approaches [44] were applied to estimate the weighted median and the mode of the density distribution of the SNP estimates. Both methods are less sensitive to the presence of potentially invalid SNPs. Finally, we assessed whether pleiotropic SNPs, thus potentially violating the exclusion restriction (horizontal pleiotropy) and the independence assumptions (no confounders), were driving the association estimates. We looked up the genetic association results of bilirubin SNPs with other phenotypes in the GWAS Catalog database [45] and obtained MR estimates and performed sensitivity analyses excluding the SNPs reaching a genome-wide threshold of association with other phenotypes.

Analyses were performed stratified by sex for pancreatic and renal cell cancers, also by subtypes for lung, ovarian, and breast cancer. We did not account for multiple testing, since we had a strong prior hypothesis based on biological plausibility and applying a strict multiple testing correction would likely have been overly conservative given the non-independence of risk for many of the cancers tested [46].

Scatter plots were used to depict the genetic association on total bilirubin levels and cancer risk. All statistical analyses and plots were performed using Stata SE14 (Stata Corporation, College Station, TX, USA) and R (*MR-PRESSO, Two-Sample MR, gwasrapidd*, and *ggplot2*; The R project). All statistical tests were two-tailed. 

## 3. Results

The minimum odds ratios (OR) that our analyses were able to detect for the *UGT1A1* SNP and non-*UGT1A1* SNPs, respectively, and each cancer are shown in Table 1. 

Each standard deviation (SD ≈ 4.4 µmol/L) increment in bilirubin levels predicted by rs6431625 in the *UGT1A1* gene was not associated with risk of pancreatic cancer (OR per one-standard deviation, 1-SD 1.02; 95% of confidence interval (CI) 0.95–1.11), whereas higher bilirubin levels predicted by non-*UGT1A1* SNPs showed an inverse association with pancreatic cancer overall (OR 0.74; 95% CI 0.61–0.89), with similar risk estimates among men and women (OR 0.75; 95% CI 0.58–0.96, and OR 0.73; 95% CI 0.55–0.97, per 1-SD increment, respectively) (Figure 1A). However, after removing SNPs (n = 22) with potential pleiotropy, these associations were attenuated toward the null (Appendix A). The scatter plot depicting the genetic associations of these instruments with bilirubin levels and with the risk of pancreatic cancer overall, and among men and women, including the likelihood-based MR estimate and its 95% CI, are shown in Appendix A. 

Neither the *UGT1A1* SNP nor the non-*UGT1A1* SNPs were associated with risk of renal cell cancer in men and women combined (Figure 1A and Appendix A); however, a suggestive inverse association was observed between bilirubin levels predicted by the non-*UGT1A1* SNPs and renal cell cancer in men (OR 0.76; 95% CI 0.57–1.00). After removing potential pleiotropic SNPs (n = 22), these associations were attenuated toward the null (Appendix A). 

Overall, we did not observe an association between bilirubin levels predicted by either the *UGT1A1* SNP or the non-*UGT1A1* SNPs and lung cancer risk (OR per 1-SD 0.98; 95% CI 0.94–1.02, by *UGT1A1* SNP; and OR 0.91; 95% CI 0.83–1.00, by non-*UGT1A1* SNPs) (Figure 1B). In stratified analyses, genetically raised bilirubin levels predicted by the non-*UGT1A1* SNPs, but not the *UGT1A1* SNP, were inversely associated with lung cancer risk among individuals who ever smoked, squamous cell, and small cell lung cancer subtypes with ORs equal to 0.86; 95% CI 0.76–0.96, 0.85; 95% CI 0.73–0.99, and 0.77; 95% CI 0.61–0.97, per 1-SD increment, respectively (Figure 1B). The scatter plots for bilirubin levels and lung cancer risk are shown in Appendix A. Among squamous cell carcinoma, the inverse associations were robust in terms of effect size to sensitivity analyses and after removing SNPs (n = 22) with potential pleiotropy (Appendix A).

Higher bilirubin levels predicted by the *UGT1A1* SNP were weakly inversely associated with risk of ovarian cancer overall and serous ovarian cancer with ORs per 1-SD increment equal to 0.96 (95% CI 0.92–1.00) and 0.94 (95% CI 0.90–0.98), respectively. In contrast, we observed a positive association between bilirubin levels predicted by non-*UGT1A1* SNPs and risk of ovarian cancer overall and serous ovarian cancer with ORs per 1-SD increment equal to 1.10 (95% CI 1.00–1.21) and 1.12 (95% CI 1.00–1.24), respectively (Figure 1C, Appendix A). However, these associations were attenuated toward the null after removing SNPs (n = 22) with potential pleiotropy. Furthermore, the MR-Egger Simex approach did not confirm the positive association suggested by the non-*UGT1A1* SNPs (Appendix A). 

Suggestive positive associations were observed between bilirubin levels, genetically predicted by non-*UGT1A1* SNPs, and risk of breast cancer (Figure 1C). 

There was little evidence for associations between genetically raised bilirubin levels by either instrument and cancers of the endometrium (Figure 1C and Appendix A) or prostate (Figure 1C and Appendix A). 

Finally, higher bilirubin levels predicted by the non-*UGT1A1* SNPs were inversely associated with risk of Hodgkin’s lymphoma (OR 0.64, 95% CI 0.42–0.99) (Figure 1D and Appendix A), while null results were observed for melanoma or neuroblastoma risk (Figure 1D and Appendix A).

### Sensitivity Analyses

The IVW risk estimates performed almost identical to the main likelihood-based risk estimates, as both methods rely on the same assumptions and suffer similarly from pleiotropy. Outlier SNPs were detected by the MR-PRESSO test in some analyses; however, their presence was not biasing the initially estimated risk effects (*p*_Distortion_ > 0.25) (Appendix A). Additionally, the MR-Egger intercept test detected overall directional pleiotropy only in the case of endometrial cancer (*p*_intercept_ = 4 × 10^−4^) and returned a potential positive association between bilirubin levels, predicted by non-*UGT1A1* SNPs, and endometrial cancer (OR 1.37; 95% CI 0.99–1.89) (Figure 1C and Appendix A). The weighted median and modal-based approaches provided similar risk estimates as the MR-Egger test in the case of endometrial cancer and as the likelihood-based MR method for the other tested cancer types (Appendix A). Finally, we identified a group of bilirubin SNPs that were genome-wide associated with other phenotypes, such as educational attainment, body mass index, and mean corpuscular volume of red blood cells. The MR analyses after removing these SNPs with potential pleiotropy (n = 20 to 22 depending on GWAS data for specific cancer types) attenuated associations of most of the ten cancers investigated, except for squamous cell lung cancer (OR 0.80, 95% CI 0.65–0.99), breast cancer (OR 1.12, 95% CI 1.04–1.20), and Hodgkin’s lymphoma (OR 0.61, 95% CI 0.32–1.14) (Appendix A).

## 4. Discussion

In this hypothesis-driven two-sample MR study, we investigated potential causal associations between genetically raised circulating bilirubin levels, a purported endogenous antioxidant, and risk of cancers of the pancreas, renal cell, endometrium, ovary, breast, prostate, lung, Hodgkin’s lymphoma, melanoma, and neuroblastoma. We found that genetically raised bilirubin concentrations were inversely associated with risk of squamous cell lung cancer and Hodgkin’s lymphoma, which is compatible with the antioxidant hypothesis of bilirubin, but positively associated with risk of breast cancer. 

### 4.1. Lung Cancer

The observed inverse association between genetically raised bilirubin levels and lung cancer risk among ever smokers, but not among never smokers, in our study are congruent with a prospective study in the UKB [14]. Similar inverse associations were also observed between serum bilirubin levels and the risk of lung cancer among male smoker in a Korean cohort (*N* cases = 240) [47] and in a prospective cohort in US (*N* cases = 386) [48]. Furthermore, in our study, we observed a robust inverse association with risk of squamous cell lung cancer subtype (OR 0.80, 95% CI 0.65–0.99 after excluding SNPs with potential pleiotropy), which is known to be strongly related to smoking. Taken together, genetically raised bilirubin levels may confer an advantage in terms of protecting people exposed to smoke oxidants against lung cancer [14,49].

### 4.2. Hodgkin’s Lymphoma

We observed robust inverse associations between genetically raised bilirubin levels and risk of Hodgkin’s lymphoma.

Our study is the first linking bilirubin metabolism to Hodgkin’s lymphoma. Given that one of the hypothesized root causes of Hodgkin’s lymphoma is infection by Epstein–Barr virus [50], bilirubin might play a role in inhibiting replication of the virus as suggested for hepatitis C virus [51] and/or by balancing oxidative stress induced by Epstein–Barr virus infection [52]. 

### 4.3. Breast Cancer and Other Hormone-Related Cancers

There was a suggestive positive association between bilirubin levels, genetically predicted by non-*UGT1A1* SNPs, and risk of breast cancer, in particular of the ER positive subtype (OR 1.12, 95% CI 1.03–1.22 after removing potential pleiotropic SNPs). These associations were relatively robust to a range of sensitivity analyses (Appendix A). A meta-analysis of retrospective case-control studies (*N* cases = 5746, *N* controls = 8365) suggested that the *UGT1A1**28 allele 7/7 genotype is a potential risk factor for breast cancer in Caucasians [53]. Given that higher circulating levels of bilirubin can inhibit glucuronidation of estrogens (estradiol) into water-soluble molecules for excretion [54] suggests that higher bilirubin may affect these hormone-related cancers indirectly by interacting with the estrogen metabolism pathway.

In contrast to breast cancer, bilirubin levels predicted by the *UGT1A1* SNP were inversely associated with risk of ovarian cancer overall and serous ovarian cancer. If the inverse associations were genuine, then these findings are potentially consistent with a second-line antioxidant defense of raised bilirubin levels in the epithelial lining of the ovaries. Similar to lung cancer, oxidative stress is a critical factor in the initiation and development of ovarian cancer [55]. We are not aware of other studies investigating either circulating bilirubin levels or the *UGT1A1* polymorphism in relation to ovarian cancer risk, and further studies are warranted.

### 4.4. Pancreatic Cancer

The findings for pancreatic cancer resemble the results of our previous two-sample MR study on the role of genetically raised bilirubin levels in colorectal cancer (CRC) using the same set of SNPs as instrumental variables [56]. We showed that bilirubin levels predicted by instrumental variables excluding the *UGT1A1* SNP were inversely associated with risk of CRC, supporting our hypothesis of anti-oxidative and anti-inflammatory properties of bilirubin. However, among men, bilirubin levels predicted by the *UGT1A1* SNP were positively associated with risk of CRC, and we argued that this could indicate either pleiotropic effects but potentially also pro-oxidative effects of an elevated bilirubin distribution among individuals with GS. 

### 4.5. Other Cancers (Renal Cell Cancer, Prostate Cancer, Melanoma, and Neuroblastoma)

We found no strong evidence for an association between genetically raised bilirubin levels and the risk of renal cell cancer, prostate cancer, melanoma, and neuroblastoma. 

Our study has several strengths. First, we used thousands of cases and controls from several large genetic consortia and published GWAS, which signified the largest and most comprehensive MR study on genetically raised bilirubin levels and cancer risk. Second, our MR assumptions were met, which were supported by our sensitivity methods and pleiotropic SNPs exclusion. Finally, common sources of bias in observational studies, including residual confounding and reverse causation, were likely reduced. This study has some limitations, first, a potentially under-powered sample size for the non-*UGT1A1* instruments to detect small effects in some cancers. For these cancers, associations can occur by chance, especially when using weak instruments and small samples, which is a phenomenon known as weak instruments bias [57]. Second, we cannot completely rule out chance in explaining the weaker observed associations. However, we had a strong prior hypothesis based on biological plausibility. Third, we also stress that the genetic instruments for bilirubin do not necessarily reflect life-long exposure. However, assuming that the association between the instruments and bilirubin is constant over time, then the MR estimate could be interpreted as an estimate of the averaged cumulative effect of bilirubin on cancer within the age range at inclusion. Fourth, although we applied several strategies to account for horizontal pleiotropy, we cannot test and exclude the possibility that the main *UGT1A1* SNP affects cancer risk through pathways other than elevated bilirubin levels. There is a large region of linkage disequilibrium across the *UGT1A* locus that includes functional polymorphisms in UGT [58,59], which aside from bilirubin, also metabolize several xenobiotic and endogenous substances including (e.g., heterocyclic aromatic amines, in well-done red meat) [56] or sex hormones [58]. Therefore, the observed differential associations of the *UGT1A1* SNP across cancer types may also reflect the modulated metabolism of such substances with carcinogenic potential. We also acknowledge that GWAS of associations of genetic variants on chronic diseases can be prone to selection bias from surviving competing risk. Methods to assess genetic effects on chronic diseases are needed to account for competing risk before recruitment [60].

Finally, we were not able to analyze non-linear associations, which would necessitate individual-level data. However, a non-linear relationship between bilirubin levels and cancer was not previously observed [56]. 

## 5. Conclusions

Genetically raised bilirubin levels were inversely associated with risk of squamous cell lung cancer as well as Hodgkin’s lymphoma, and positively association with risk of breast cancer. These findings should help in setting priorities in future research on bilirubin levels and cancer risk. If confirmed in other studies, bilirubin could be a promising marker to risk stratify individuals for more frequent screening of selected cancers. 

## Figures and Tables

**Figure 1 cells-10-00394-f001:**
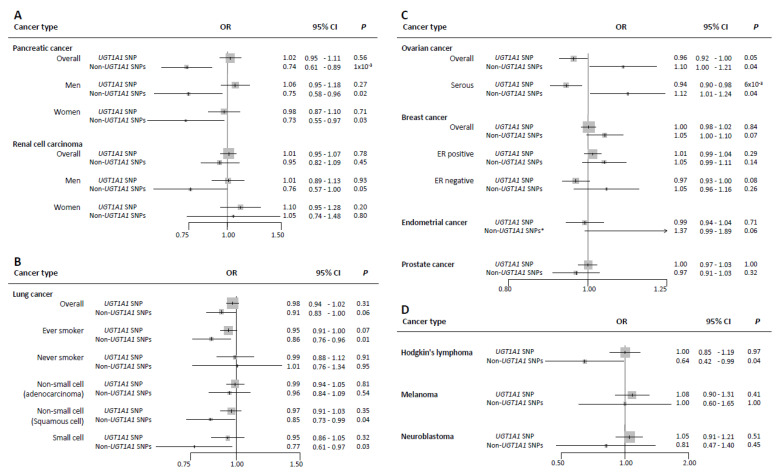
Forest plot of associations between genetically-predicted bilirubin levels and risk of ten cancers (per one-standard deviation (1-SD) increment in circulating total bilirubin levels equivalent to ≈4.4 µmol/L). (**A**): The association between 1-SD increment in bilirubin levels predicted by *UGT1A1* SNP and the non-*UGT1A1* SNPs with risks of pancreatic cancer and renal cell carcinoma (overall, men, and women). (**B**): The association between 1-SD increment in bilirubin levels predicted by *UGT1A1* SNP and the non-*UGT1A1* SNPs with risk of lung cancer (overall, ever smoker, never smoker, adenocarcinoma, squamous cell, and small cell lung cancers. (**C**): The association between 1-SD increment in bilirubin levels predicted by *UGT1A1* SNP and the non-*UGT1A1* SNPs with risks of ovarian cancer (overall and serous), breast cancer (overall, ER positive, and ER negative), endometrial cancer, and prostate cancer. (**D**): The association between 1-SD increment in bilirubin levels predicted by *UGT1A1* SNP and the non-*UGT1A1* SNPs with risks of Hodgkin’s lymphoma, melanoma, and neuroblastoma. The results are provided by a likelihood-based MR test, * with the exception of endometrial cancer, which results are provided by the Egger-MR test. Abbreviations: OR odds ratio, CI confidence interval, *p p*-value, ER estrogen receptor.

**Table 1 cells-10-00394-t001:** Summary information of cancer GWAS samples and power assessment.

Cancer Type	Subtype	*N* Cases	*N* Controls	SNP Set	Minimum Detectable OR
Pancreatic cancer	overall	7110	7264	*UGT1A1* SNP	1.12/0.89
				Non-*UGT1A1* SNPs (n = 113)	1.30/0.77
	men	3861	4056	*UGT1A1* SNP	1.17/0.86
				Non-*UGT1A1* SNPs (n = 113)	1.43/0.70
	women	3252	3268	*UGT1A1* SNP	1.18/0.84
				Non-*UGT1A1* SNPs (n = 113)	1.48/0.67
Renal cell cancer	overall	10,784	20,406	*UGT1A1* SNP	1.08/0.92
				Non-*UGT1A1* SNPs (n = 111)	1.21/0.83
	men	3227	4916	*UGT1A1* SNP	1.17/0.86
				Non-*UGT1A1* SNPs (n = 109)	1.43/0.70
	women	1992	3095	*UGT1A1* SNP	1.22/0.82
				Non-*UGT1A1* SNPs (n = 109)	1.58/0.63
Lung cancer	overall	29,266	56,450	*UGT1A1* SNP	1.05/0.95
				Non-*UGT1A1* SNPs (n = 109)	1.12/0.89
	ever smokers	23,223	16,964	*UGT1A1* SNP	1.07/0.93
				Non-*UGT1A1* SNPs (n = 109)	1.17/0.85
	never smokers	2355	7504	*UGT1A1* SNP	1.17/0.85
				Non-*UGT1A1* SNPs (n = 109)	1.46/0.69
	adenocarcinoma	11,273	55,483	*UGT1A1* SNP	1.07/0.93
				Non-*UGT1A1* SNPs (n = 109)	1.18/0.85
	squamous cell	7426	55,627	*UGT1A1* SNP	1.09/0.92
				Non-*UGT1A1* SNPs (n = 109)	1.22/0.82
	small cell	2664	21,444	*UGT1A1* SNP	1.15/0.87
				Non-*UGT1A1* SNPs (n = 109)	1.39/0.72
Ovarian cancer	overall	25,509	40,941	*UGT1A1* SNP	1.06/0.95
				Non-*UGT1A1* SNPs (n = 111)	1.14/0.88
	serous	16,003	40,941	UGT1A1 SNP	1.07/0.94
				Non-*UGT1A1* SNPs (n = 111)	1.16/0.86
Breast cancer	overall	122,977	105,974	*UGT1A1* SNP	1.03/0.97
				Non-*UGT1A1* SNPs (n = 112)	1.07/0.94
Endometrial cancer	overall	12,906	108,979	*UGT1A1* SNP	1.07/0.94
				Non-*UGT1A1* SNPs (n = 110)	1.16/0.86
Prostate cancer	overall	79,194	61,112	*UGT1A1* SNP	1.04/0.96
				Non-*UGT1A1* SNPs (n = 107)	1.09/0.92
Hodgkin’s lymphoma	overall	1200	6417	*UGT1A1* SNP	1.24/0.81
				Non-*UGT1A1* SNPs (n = 91)	1.65/0.61
Melanoma	overall	1804	1026	*UGT1A1* SNP	1.31/0.77
				Non-*UGT1A1* SNPs (n = 75)	1.86/0.54
Neuroblastoma	overall	1627	3254	*UGT1A1* SNP	1.23/0.81
				Non-*UGT1A1* SNPs (n = 57)	1.62/0.62

Abbreviations: *N* Number, OR odds ratio, SNP single nucleotide polymorphism

## Data Availability

The data presented in this study are available on request from the corresponding author freislingh@iarc.fr. Requests for the data require formal approval by the principal investigators of each genetic consortium.

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
