# Peer review of "Genetically Raised Circulating Bilirubin Levels and Risk of Ten Cancers: A Mendelian Randomization Study"

_cells, 2021, doi:10.3390/cells10020394_

Round 1

Reviewer 1 Report

The manuscript reports an interesting and relevant study on the potential causal relationship between bilirubin – a purported endogenous antioxidant - and several cancers using a two-sample Mendelian randomisation approach. It is written clearly, and the results are nicely presented. The methods also seem sound and I only have some minor comments:

  • Line 93: I am not clear on what is meant by “data availability” in terms of outcome selection. Are these the only cancers with published GWAS summary statistics? I would normally expect outcomes to be selected based on biological plausibility unless it is intended as a negative control outcome. 
  • Line 120: It would be helpful to know whether these cases were mostly incident cancers, prevalent (i.e. cancer survivors) or a mixture. There is always a potential for selection bias but this could be particularly problematic if many of the cases are cancer survivors who gave informed consent post-disease rather than incident cancer cases. For example, life-threatening ADRs to chemotherapy due to UGT1A1 deficiency (also an important drug metabolising enzyme) could distort genotype frequencies if survival is also affected. Ideally, a sensitivity analysis on case ascertainment would be desirable excluding those study with a high potential for bias. However, I suspect few of these GWAS use cohort data and some further acknowledgement of the potential for selection bias in the discussion would suffice.
  • Line 208: I think there is a “by UGT1A1 SNP” missing from the brackets.
  • Line 320: I would argue it is too strong to say these biases are “avoided” because this relies on some strong assumptions. There could still be residual confounding if ethnicity has not been deeply captured across all GWAS’. The main promoter variant is in LD with several other functional SNPs that may “confound” some of the cancers particularly where antioxidant activity by bilirubin is likely to have a weaker protective role (see point below).
  • The discussion could benefit from the acknowledgement of the large region of linkage disequilibrium across the UGT1A locus that includes UGT1A1*28 and several other functional polymorphisms (e.g. ugt1a6). These could impact the glucuronidation activity toward several endo/exogenous substances (hormones/carcinogens etc) aside from bilirubin and have a differential effect on cancer types. https://clincancerres.aacrjournals.org/content/11/3/1348. Probably not relevant for lung cancer but important when considering several types of cancer including those where hormones have a role.

Reviewer 2 Report

This research paper would be very intersting.

In the abstract, line 52, you mentioned the different range of OR as results for SNPs, I should suggest that you might indicate the value of p, because one of your OR intervals as results is close to the 1 value. 

Also, I recommend reviewing the anti-inflammatory properties (line 71) of bilirubin and adding another possible references that corroborate this more clearly (not only one, ref.4). You can check: 

  • Lee Y, Kim H, Kang S, Lee J, Park J, Jon S. Bilirubin Nanoparticles as a Nanomedicine for Anti-inflammation Therapy. Angew Chem Int Ed Engl. 2016 Jun 20;55(26):7460-3. doi: 10.1002/anie.201602525. Epub 2016 May 4. PMID: 27144463.
  • Tsai MT, Tarng DC. Beyond a Measure of Liver Function-Bilirubin Acts as a Potential Cardiovascular Protector in Chronic Kidney Disease Patients. Int J Mol Sci. 2018 Dec 29;20(1):117. doi: 10.3390/ijms20010117. PMID: 30597982; PMCID: PMC6337523.
  • Bock KW. Human AHR functions in vascular tissue: Pro- and anti-inflammatory responses of AHR agonists in atherosclerosis. Biochem Pharmacol. 2019 Jan;159:116-120. doi: 10.1016/j.bcp.2018.11.021. Epub 2018 Dec 1. PMID: 30508524.

In material and methods, you described the two-sample MR approach. In this way, line 102, you mentioned the raw total bilirubin levels were adjusted for age, sex, 40-top components-stratification... but you didn't write anything about ethnic variables or more information that can better define the population being analyzed. Maybe you were minded with this but it has not been reflected in the paper. It is should also be present for genetic testing if an European ancestry has been used as controls.

To see: 

  • Hansen TWR, Wong RJ, Stevenson DK. Molecular Physiology and Pathophysiology of Bilirubin Handling by the Blood, Liver, Intestine, and Brain in the Newborn. Physiol Rev. 2020 Jul 1;100(3):1291-1346. doi: 10.1152/physrev.00004.2019. PMID: 32401177.
  • Wang L, Yang CX, Men X, Dong XM. [Analysis of the influencing factors of serum bilirubin in workers exposed to occupational hazards factors in a urban area]. Zhonghua Lao Dong Wei Sheng Zhi Ye Bing Za Zhi. 2020 Dec 20;38(12):894-897. Chinese. doi: 10.3760/cma.j.cn121094-20190826-00353. PMID: 33406546.
  • Maya-Enero S, Candel-Pau J, Garcia-Garcia J, Giménez-Arnau AM, López-Vílchez MÁ. Validation of a neonatal skin color scale. Eur J Pediatr. 2020 Sep;179(9):1403-1411. doi: 10.1007/s00431-020-03623-6. Epub 2020 Mar 10. PMID: 32157460.

In limitations (line 321), you commented some different aspects to take care, maybe one of this could be the diet and pathways to related bilirubin levels and some foods and dietstyle or pharmaceutic specific treatments?

Maybe conclusions could be improve with more details about what benefits we (society) can find if we use bilirubin levels as a specific cancer biomarker in the future.

Reviewer 3 Report

The authors present an interesting study on the SNPs related to the bilirubin level and the risk of 10 different types of cancers. The paper is well written, describing in depth the methods, the results, and discussing the most important results corroborated with other studies.

There are some minor changes that could be done:

  • the abstract is better to have the structure: Background, Methods, Results, Conclusions
  • some editing is needed for the subtitles in Methods, Results, or Discussions. It seems that these subtitles can be left out as the text is clear enough.
  • The paragraph from line 315 can be included with that from line 322 in the Strengths and Limitations of the study. I would emphasize also the strengths not only the limitations
  • A check for all the manuscript for typos is needed.
